# Challenging the Fundamental Premise of White Supremacy: DNA Documents the Jewish Origins of the New England Colony

Elizabeth C. Hirschman

Department of Business and Economics, University of Virginia-Wise, Wise, VA 24293, USA; elizabeth524@aol.com

**Abstract:** The English Puritans of New England are a foundational element in the current racist ideology of White Supremacy. Depicted in history books as stalwart British Protestants who braved bitter winters and Native predations to establish a "City on the Hill"—a beacon to the world of freedom and liberty—the Puritans became ideals in the American consciousness. But what if this is a misrepresentation, created largely in the mid and late 1800s to serve as a political barrier against Catholic, East European, Jewish, and Asian immigrants who threatened the "American way of life"? The present research uses genealogical DNA data collected from descendants of the New England settlers to demonstrate that these original "Yankees" were of Jewish ancestry. The WASP origination of New England is shown to be a false narrative.

**Keywords:** New England; genealogical DNA; Ashkenazic Jews; white supremacy

## 1. Introduction

New England. These two words immediately conjure up images of colonial Yankee villages with stone houses, snow on the ground, and church bells peeling in the crisp, wintry air. Families of pious Puritans make their way to Sunday services. These, we have been taught, were the white, Anglo-Saxon Protestant pioneers who created the extraordinary economic and political success of the United States, the originators of American cultural supremacy.

In his preface to the Chronicles of the First Planters of the Colony of Massachusetts Bay 1623–1636 (Young [1846] 2015) exemplifies the hagiographic writing style used to lionize these early colonists: "No nation or state has a nobler origin or lineage than Massachusetts. My reverence for the character of its founders constantly rises with the close study of their lives, and a clearer insight into their principles and motives . . . Their worth has never been over rated and we should never tire of recounting their virtues".

But what if this image (Figure 1) is a fiction—an amalgam of racial ideologies grounded the false belief that North America was colonized by British Protestants of Anglo-Saxon ancestry. What if, in fact, the settlers of New England were not who we thought they were? What if they were actually Ashkenazic (Eastern European) Jews who were not only not Protestant, but not even ethnically British, and not even "white" as that term is usually applied? What happens then to the doctrine of White Supremacy?

Recent DNA research on the descendants of the colonial settlers of Central Appalachia (Hirschman et al. 2019a), the Mayflower/Plymouth colony (Hirschman et al. 2018), the Roanoke Colony (Hirschman et al. 2019b), and the Post-Civil War outlaws Jesse and Frank James, the Cole Younger gang, and the Hatfield Clan of West Virginia (Hirschman and Vance 2021) have shown that these persons were of Sephardic and Ashkenazic Jewish ancestry, supplemented with Southeastern Europe and Southern India ancestry. The present study extends these findings to men in the Colonial Massachusetts colony.

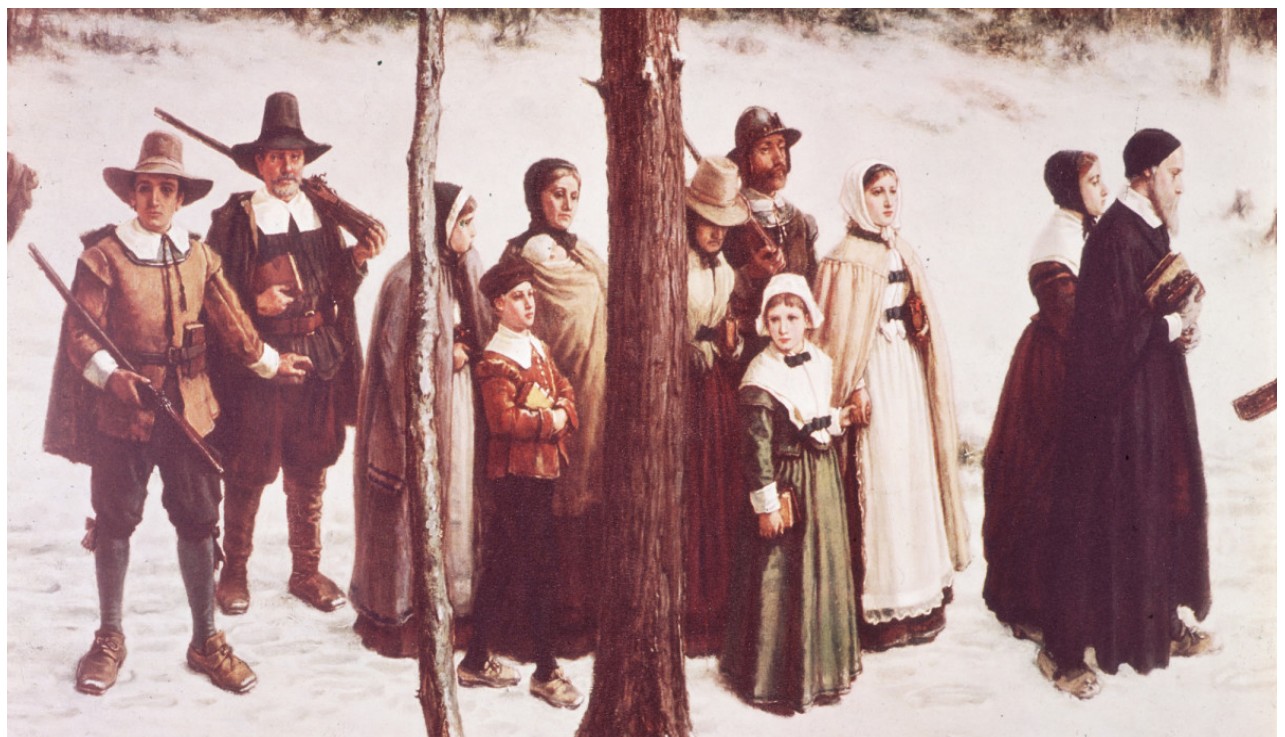

**Figure 1.** Painting of Puritans on their way to Church.

### 1.1. Writings on White Supremacy

One of the most influential American writers arguing for White Supremacy was Madison Grant (1921, [1933] 2018). Grant held several important quasi-academic titles that gave his works credibility to the public: President of the New York Zoological Society, Trustee of the American Museum of Natural History, President of the Boone and Crockett Club, and Councilor of the American Geographical Society. His most cited work, *Passing of the Great* Race (1916), put forward the argument that North Western Europeans (Nordics) were the evolutionary epitome of the human species, but had been weakened over time by incursions from lesser races, such as Eastern Europeans, Asians, and Mediterraneans.

In a second book, *The Conquest of a Continent* (1933), Grant praised the racial superiority of the "Nordics" who he believed composed the early settlers in North America; these primarily included the "Anglo-Saxon" inhabitants of England who, Grant stated, formed the front ranks of New England colonists. As he puts it (Grant [1933] 2018, p. 66), "The New England Puritans . . . were pure English from the most Anglo-Saxon part of England and consisted largely of yeomen and lesser gentry . . . They were essentially dissenters who refused to bend the knee to prelate or king".

In these sentiments, Grant embraces the contemporaneous ideology of Nazi eugenicists such as Hans Gunther (1927, p. 49) who proposed that human races could be classified and identified as to both their mental and physical characteristics: "A race shows itself in a human group which is marked off from every other human group through its own proper combination of bodily and mental characteristics, and in turn produces only its like". Thus, within this framework, Nordics remain Nordics and Jews remain Jews regardless of what country they may reside in. White Supremacy is—at its core—a theory of genetic ethnic differentiation. One's genes carry one's ethnicity, not one's language or country of origin.

In the Introduction to Grant's second book (Grant [1933] 2018), a colleague, Henry Fairfield Osborne, states, "The character of a country depends upon the racial character of the men and women who dominate it. (p. vii) . . . Moral, intellectual, and spiritual traits are just as distinctive and characteristic of different races as are head form, hair and eye color, physical stature, and other data of anthropologists (viii) . . . The theme of (Grant's present work) is that America was made by Protestants of Nordic Origin and that their ideas about

what makes true greatness should be perpetuated … This is a precious heritage which we should not impair or dilute by permitting the entrance and dominance of alien values and people of alien minds and hearts (p. ix)".

Grant, himself, (Grant [1933] 2018, p. 2) presents this thesis clearly, "In the days of our fathers, the white population of the United States was practically homogeneous. Racially, it was preponderantly English and Nordic. Our institutions remained overwhelmingly Anglo-Saxon down to the time of the Civil War … In America the events of the last hundred years have greatly impaired our purity of race and our unity of religion and even threatened our inheritance of English speech (p. 5)".

In these writings, we can discern the genesis of the current social and political upheaval over immigration and racial equity (e.g., the 6 January 2021 attack on the US Capitol) as a continuation of assumptions about the ethnic origins of the United States (Belew 2018; Chalfant 2019; Chow 2018; Daniels 1997; Dobratz and Shanks-Meile 2000, 2006; Gillborn 2006; Sands 2019). The research we report here shows that this assumption—that New England was settled by Anglo-Saxon/Nordic persons—is false.

### 1.2. Who Were the Puritans, Really?

Efforts to glorify the British ancestral origins of the New England Puritan settlers were well underway by the mid-1850s in conjunction with the arrival of Irish Catholic immigrants in New York and Boston (Brown 2010; Elson 1964). These efforts intensified during the early 1900s, as Italian, Greek, Polish, and Ashkenazic Jewish immigrants passed through Ellis Island into the Northeast (see e.g., Stoddard [1920] 1921, 1922). In order to create a social distinction between the earlier New England colonists and these undesirable newcomers, family genealogy books were written purportedly documenting the noble and aristocratic pedigrees of the original settlers (see e.g., Smith [1897] 2015). These books were intended to document that the writer and his/her relatives were of unimpeachable English ancestry. However, upon closer examination, a contrary subtext becomes visible. A good example is the family history written by Frederick William Gookin ([1912] 2015) about his Puritan ancestors who helped settle Massachusetts.

The book's author identifies his English ancestors as being from Canterbury, Kent, noting that his earliest ancestor William Cokyn/Gokin founded a hospital there in 1199 (Gookin [1912] 2015, p. 1). He tracks the family through Kent history to the late 1500s and reports a close cross-generational marital pattern between the Gookin and Denne families. He writes that the Denne/Dene family arrived in England in 1066 from Normandy, France, with William the Conqueror (Gookin [1912] 2015, p. 8). At this point, we call attention to the fact that the surname Denny/Dene can be a French form of the Hebrew name Daniel, and we will later show that this family is of Ashkenazic Jewish descent. Gookin ([1912] 2015) further states that the Denne family married into the Coppin family of Kent (Gookin [1912] 2015, p. 9), and we interject here that Coppin is a medieval form of the Hebrew name Jacobs ([1887] 1893).

Additional investigation of the Cokyn/Gooken surname reveals that the word *gooken* is German slang for "watcher or looker": i.e., *Gucken,* along with its spelling variant kucken, is the common word for "to look" in northern and central Germany. It is also Yiddish (i.e., Ashkenazi Jewish German) in the form Guggen, as in Guggenheim. Thus, there are some important initial clues about the ethnic origins of these inhabitants of Canterbury, England, many of whom pledged themselves to Puritanism during the late 1500s and ventured to North America.

### 1.3. Jews Arrive in England with William I, the Conqueror

One may ask when and how Ashkenazi Jews, who we usually associate with Eastern Europe, could have arrived in England? The answer lies in the Norman Conquest of 1066. Documents dating from this period show that William I brought large numbers of Ashkenazi Jews from Normandy in Northern France near the Alsace Lorraine Region into England during and after his conquest of that country. According to some records, he

believed Jews were needed to serve as bankers and financial agents so that he could enact his ambitious post-Conquest building program. Christians, at that time, were not allowed to engage in usury, and William may have had pre-existing relationships with the large Ashkenazi community in Northern France (see, e.g., Golb 1998; Jacobs [1887] 1893; Samuel 2004).

Over the next few centuries, this led to several Jewish families becoming large landowners and government administrators, often with noble titles (Jacobs [1887] 1893). Some Jewish bankers—such as Aaron of Lincoln and Licoricia of Oxford—became extremely rich. However, most of the incoming Jews filled a range of occupations within their English communities as doctors, goldsmiths, silversmiths, vintners, cloth merchants, and fishmongers. They lived in many parts of England and Wales, sometimes in distinct communities practicing their rituals, but often alongside the wider population (Jacobs [1887] 1893). Jews were allowed to travel freely across England and Wales and were granted a "Charter of Liberties" by King Henry I, meaning they could shelter in any of the King's castles if they were in danger (Jacobs [1887] 1893).

However, some influential nobles became deeply indebted to the Jews from whom they had borrowed funds and, not desiring to repay these loans, they pressured Edward I to revoke the Jews' charter. In July 1290, Edward I issued an Edict of Expulsion requiring all practicing Jews to "depart from England". It is known that many of the most well-connected Jews at that time chose to convert (at least publicly) to Christianity, while retaining their Jewish practices in the home (Jacobs [1887] 1893). Thus, they became what are now termed *crypto-Jews* (Hordes 2005; Israel 2002; Kagan and Dyer 2011; Kagan and Morgan 2009). The genealogical DNA results we report document that these English crypto-Jews may have retained their ancestral identities and immigrated as a group to New England in the early 1600s. This would have meant retaining their crypto-Jewish practices for a 300 year period, which may seem unlikely, but is currently present among colonial Hispanic families in the Southwest United States (see e.g., Hordes 2005).

## 2. Methodology

### 2.1. Using DNA to Determine Ethnic Origins

This research presents genealogical evidence grounded in the most recent DNA technology to show that while the point of departure for the Massachusetts Bay Colony settlers was indeed England, their ethnic origins were in the Middle East, the Mediterranean, and Eastern Europe. Thus far, DNA data have been generally avoided among social historians who prefer to use contemporaneous documents as their primary, often sole, data source. We propose that this document-dependence can lead to the misinterpretation of historical events.

One of the current buzzwords in the social sciences is *disciplinary silos*. This concept draws attention to the insularity of academic disciplines and research approaches that use only one dominant paradigm or method to study a phenomenon. Over the past few decades, greater strides have been made within and across academic fields when ideas and ways of thinking are imported from other disciplines, rather than using only traditional approaches. We propose that the same can be said for the discipline of social history studies. Whether examining the last decade, the last century, or the last millennium, historians have largely depended upon written, carved, printed, and, more recently, electronic images to conduct their research.

By contrast, archaeologists, who also study human history, have been more open to adopting novel technologies such as satellite photography, in-ground radar, and magnetic imaging in their research. These have provided novel perspectives to these researchers, permitting them to locate, for example, the outlines of flattened city-walls, which are not visible when a site is only viewed from ground level or excavated downward.

We propose that current social historians resemble the archaeologists of old—they dig in one narrow trench using available documents and consider that an accurate excavation of an historical event. In so doing, they may miss the big picture. While historical documents

may be lost, falsified, or written to be purposely misleading, DNA is what it is—chemical molecules in each and every cell of one's body. Properly collected, analyzed, and compared to other samples, DNA does not lie.

In 2000, Family Tree DNA (FTDNA.com, accessed multiple times from 8 January to 23 April 2021) became the first company offering genealogical DNA testing for ancestry research. Since then, several other companies have begun offering these services. In 2019, new genealogical analysis tools were developed; these included auto-clusters (visually grouping persons with matching DNA markers into clusters) and family tree theories (suggesting possible relationships between DNA matches by combining several family trees as well as global phylogenetic trees). This new technology permits users to track the genetic evolution of their ancestry in a fine-grained manner; one's ancestors' paths across both time and space can now be viewed, sometimes to within a few hundred years. Presently, it is estimated that the major genealogical testing companies have accumulated about 26 million DNA profiles (FTDNA.com). Most companies have posted their test results on multiple commercial sites, giving users at each one access to all data collected globally.

### 2.2. How Do They Work?

A genealogical DNA test is performed on a sample provided by the individual. After following the kit instructions on how to collect the sample, the user returns it for analysis. The sample is then processed using a technology known as DNA microarray to obtain the genetic information requested by the consumer (Bettinger and Wayne 2016).

### 2.2.1. Male Y Chromosome (Y-DNA) Testing

In the present study, we use only male DNA samples, as the MtDNA lineages available for the Massachusetts Bay settlers had inadequate lineage tracing. The Y-Chromosome is one of the 23rd pair of human chromosomes. Only males have a Y-chromosome, because women have two X chromosomes in their 23rd pair. A man's test results may be compared to another man's results to determine the time frame in which the two individuals may have shared a most recent common ancestor (MRCA) in their direct paternal line (Bettinger and Wayne 2016). There are two types of paternal DNA testing: STR and SNP.

### 2.2.2. STR Markers

The most common type of testing is performed using STR (short tandem repeat) markers. A certain section of DNA is examined for a pattern that repeats basic chemical components of the DNA. The number of times it repeats is the value of the marker. Typical tests examine between 12 and 111 STR markers. STRs mutate fairly frequently, which permits different branches of paternal ancestry to be charted. The results of two individuals can then be compared to see how closely they are related. DNA testing companies usually provide an estimate of how closely related two people are, in terms of generations or years, based on the differences between their results (Bettinger and Wayne 2016).

A person's male ancestral haplogroup can often be inferred from STR results but can be proven only with a Y-chromosome SNP test (Y-SNP test). A single-nucleotide polymorphism (SNP) is a change to a single nucleotide in a DNA sequence. Typical Y-DNA SNP tests examine about 20,000 to 35,000 SNPs. Getting an SNP test allows a much higher resolution of one's male ancestry than STRs. It can also be used to provide additional information about the relationship between two individuals across time and to confirm one's ancestral haplogroup. In the present research, both STR and SNP Y-DNA results are used to determine the New England male settler's ethnic ancestry.

### 2.3. Sample Frame

The surname sample frame used for the study is the Winthrop Society Settler List. The website announcement for this states:

"The Puritans of the Massachusetts Bay Company and their elected Governor, John Winthrop, emigrated to New England in 1630 to found a 'City on a Hill.' The Winthrop Society: Descendants of the Great Migration is dedicated to honoring and preserving their memory, philosophy and tradition, and transmitting their example of courage, faith, civic duty and integrity". (excerpt from the Winthrop Society Charter). (www.winthropsociety. com, accessed multiple times from 8 January to 23 April 2021). We primarily relied upon Y DNA samples for these settlers, which were available in specific surname studies (e.g., Denny), because this permitted checking the markers against multiple donors.

### 3. Hypothesis

Our principal hypothesis is that the Massachusetts Bay Colony was settled by persons whose male ancestors were Ashkenazi Jews arriving in England from 1066 CE to the early 1200s CE. These persons had been living in northern (Normandy) France and migrated to England at the invitation of William I after his Conquest of England in 1066 CE. The original Ashkenazi Jewish population was formed in the Holy Roman Empire during the late 900s CE by Jews moving westward from the Holy Land, Italy (formerly the Roman Empire), and the Mediterranean Region, and the population was located primarily in northern France and Germany (Vital 1999).

During the late Middle Ages, the Ashkenazi population shifted steadily eastward into areas that became part of the Polish-Lithuania Commonwealth, comprising parts of what are now Belarus, Estonia, Latvia, Lithuania, Moldova, Poland, Russia, Slovakia, and Ukraine (Vital 1999). Many of these Eastern European Ashkenazim immigrated to the United States during the late 19th and early 20th centuries (Vital 1999). Their descendants comprise the Jews described by Madison Grant ([1933] 2018, p. 227) as follows: "The (Ashkenazic Jews) set up almost entirely in the larger cities (and) built up a ghetto population similar to the . . . congested urbanism of their homeland . . . Americans were so obsessed with the idea of providing a "refuge for the oppressed" that they welcomed into our country that morass of human misery found in the Polish ghettoes!"

This makes clear the anti-Semitism found in the White Supremacist ideology of Grant's era—which became horrifically enacted during the Third Reich—and is carried forward today among White Supremacists (Longerich 2010; Fredrickson 1981; Rosenberg 1930; Winant 1997). We will show that the DNA genetic profile of the Jewish "morass of human misery" Grant decries is the same as the DNA genetic profile he celebrates in the Massachusetts Bay Colony.

### 4. Analysis and Findings

Our Y-DNA samples are drawn from the members of the Colonial Massachusetts Bay Winthrop Society, whose paternal ancestors could be located within a publicly accessible surname database; this was done to permit other researchers to access the same data and check the results reported here. Each surname is discussed in alphabetical order.

#### 4.1. Abbott

The DNA profile of George Abbott of Andover, Mass., is given below. It belongs to the E-m35 haplogroup. This haplogroup originated in the Middle East and is considered one of the founding ancestries of the Jewish people (www.jewishgen.org, accessed multiple times from 8 January to 23 April 2021). Thus, it is highly likely that this colonist had Jewish (or at least Semitic) ancestors.

George Abbott

1 B108065 Abbott  George E Abbott III of Andover MA b 1615 d 1681  England E-BY65664 13 24 13 10 16-17 11 12 12 13 11 31 19 9-10 11 11 26 14 20 31  14-16-17-17 9 11 19-21 15 12 16 20 32-34 13 12 14  11 11

Using the phylogenetic tree below (Figure 2), we can trace George Abbott III's ancestry (E-V13) from its origin in the Middle East into Croatia and Bulgaria some 2800–2500 years ago.

E-Z1919 Z1920/CTS4235/PF2228 * CTS202/Z825/V1083 * Y2478/S8336/V1084+2 SNPs-formed 13400 ybp, TMRCA 12100

- E-V22 CTS5295 * CTS2817 * Y2528/FGC7782+60 SNPsformed 12100 ybp, TMRCA 8500 ybpinfo
  - E-V22*
    **JK2888 Ptolemaic Egypt, Abusir, 2100-2000 ybp (97-2 BCE)**
- E-L618 Y3763/FGC11427 * CTS6178 * CTS10912/PF2249+49 SNPsformed 12100 ybp, TMRCA 7600 ybpinfo
  - E-L618*
    **I3948 Cardial Neolithic, Zemunica Cave Croatia 8000-7800 ybp**
  - E-V13 V13/PF2211 * L1024/CTS3726/PF2226 * PF2222/Z1051+34 SNPsformed 7600 ybp, TMRCA 5100 ybpinfo
    - E-V13*
      **P192-1 Thracian, Svilengrad Bulgaria 2800-2500 ybp (800-500 BCE)**

**Figure 2.** Phylogenetic Tree of E-M35.

*4.2. Abel*

The Abel DNA profile below was a 22 out of 24 marker match with a man named Christopher Hunt, 1728–1781, in England, whose descendant is a member of the Iberian–Ashkenazi DNA Project (FTDNA). That project includes men who know they are of Ashkenazic Jewish ancestry, but whose ancestors sojourned for some time in Spain. This indicates that the Abel ancestor was likely an Ashkenazic Jew. It is worth noting, as well, that the surname Abel—just as with Adams, which will be discussed below—is a Hebrew name.

| 170296 | England | I-M253 | 15 22 14 10 | 13-14 | 11 14 11 12 11 28 15 8-9 8 11 21 16 19 28 | 12-15-15-15 | 10 10 | 19-21 | 14 15 16 19 | 35-36 | 1 |

*4.3. Henry Adams: The Presidential Adams Lineage*

Because of their famous ancestral lineage, the Adams DNA Project (FTDNA.com) has markers from several descendants of Henry Adams, the Adams family progenitor from England. Two of these participants have had advanced testing of their haplotype. These indicate that the Presidential Adams haplotype is in group R-S14328 and subgroup R-FGC23887.

When compared to others having this same refined DNA SNP pattern, the following surnames were found in the Jewish Heritage Project. This database contains DNA entries from persons of known Jewish ancestry (Familytreedna.com. date accessed): Lorenzini (Italy), Goethals (Belgium), Kanbier (Netherlands), Koch (Germany), Merket (Germany), and Montonari (Ferrara, Italy). Some of these surnames are suggestive of Ashkenazic Jewish ancestry, and the Italian surnames also may indicate that the Adams progenitor was a very early adherent to Judaism, possibly coming from the Mediterranean or Middle Eastern region and venturing to Italy under Roman rule.

Given that the surname Adams, itself, is Hebrew in origin and that President John Adams' great grandfather married one of the Plymouth Colony descendants who was Jewish (see Allied Families (Y-DNA)—Mayflower DNA R-p312 matches for Alden), we believe a strong presumption can be made that the Adams family was ancestrally Jewish and may have been aware of that when choosing marriage partners.

169774　Adams　Henry Adams 1583 Somerset, Eng. thru son Samuel England, R-Z4471　23-23 16 10 12 12 16 8 12 21 19 14 13 11 13 11 11 12 1235149
*1325141111-1512121213132916 9-10111125152029*



### 4.4. John Adams

A second Adams family, whose markers are shown below, also immigrated to the Massachusetts Bay colony in the early 1600s. Their biographical details are given below (Table 1).

**Table 1.** Biographical details of second Adams family.

| John Adams, Sr. | |
| --- | --- |
| Also Known As: | "Arrived Plymouth Colony", "Fortune", "1622", "John the Plymouth Pilgrim" |
| Birthdate: | before 1600 |
| Birthplace: | Shropshire, England, United Kingdom |
| Death: | circa 11 November 1633 <br> Plymouth, Plymouth Colony, Colonial America |
| Place of Burial: | Plymouth, Plymouth, Massachusetts, United States |
| Immediate Family: | Son of unknown father of John Adams of the Plymouth Colony and unknown mother of John Adams of the Plymouth Colony <br> Husband of Elenor Winslow <br> Father of Susanna Adams; Capt. John Adams and James Adams, of Scituate |

This Adams family ancestor has an E-m35 DNA haplotype. As already discussed, E-m35 is recognized as a Semitic haplogroup originating in the Middle East and is a founding lineage of the Jewish people (Jewishgen.org, accessed multiple times from 8 January to 23 April 2021). Thus, this Adams family is also likely of Jewish ancestry (Hirschman 2021).

John Adams England E-m35
13 25 14 10 17-19 11 12 11 14 11 32 17 9-9 11 11

### 4.5. Agar

The DNA markers shown below were a weak match (20 out of 24 markers) to several persons in the Iberian–Ashkenazi DNA Project (FTDNA) described earlier. The matches included a John Lynch from Ireland, Robert Stiles from England (1635–1690), a Gonzales from Spain, and Jose de Villagran (b. 1670) from Mexico. The mixture suggests that these Ashkenazic Jews ventured into the British Isles and to Spain. Agar may have then emigrated from England to Massachusetts.

Ralph Agar ca 1674–1729 Beverly, Yorkshire, England R-m269
13 25 14 10 11-13 12 12 13 13 13 29 16 9-9 11 11 25 15 19 31 15 16-17-17 11 11 19-23

### 4.6. Bachiler

The Batchelor/Bachilor DNA markers below matched several persons from the Iberian–Ashkenazic Jewish database at a 20 out of 24 marker level or above. The matches included a man named Sulaiman Khalil Khashif from Sudan, Zalman Zev Shapiro (Lithuania), Charles Whalen (Ireland), Jose Luis Roble (Mexico), John Lanier (1499 England), and Israel Boone—the son of Daniel Boone. The Boone family was shown to be of Jewish descent in an earlier article (Hirschman et al. 2019a).

| | | | | | | | | | | | | | | | |
| --- | --- | --- | --- | --- | --- | --- | --- | --- | --- | --- | --- | --- | --- | --- | --- |
| Richard Batchelor | United Kingdom | R-M269 | 13 | 24 14 11 11-14 | 12 | 12 12 12 14 29 16 9-10 11 | 12 11 13 15 15 90 15-15-17-18 | 11 | 1 19-1 23 | 15 | 11 11 5 77 37-39 | 12 11 21 | 15-9 16 | 8 11 00 11 8 00 2 24 | 23-24 | 18 11 11 0224 8 12 22 11 11 11 11 13 22 03 21 41 12 25 |
| 363981 | Robert Batchelor b. 1520 | England | R-BY57400 | 12 34 14 | 11 | 11-14 11 12 22 24 9 | 17 | 9-10 12 11 15 59 | 30 | 15-15-17-18 | 11 1 | 19-23 11 55 7 | 17 | 37-39 11 22 1 | 9 | 15-16 8 00 11 8 00 12 23-24 11 11 80 22 24 8 12 22 11 11 11 11 23 03 21 41 12 2 |

### 4.7. Ballard

The Ballard DNA markers belong to a sub-group called R-L48. Persons in this sub-group are often from Friesland and associated with the Anglo-Saxon presence in England. However, more recent research, the R-U106XL48 DNA Project, has identified some specific persons who have markers very close to those of the Ballards. Their surnames are Arnold (England), Izikson (Russian Federation), and Faunt (England). Arnold is a known British–Jewish surname, and Izikson from Russia is also Jewish. Additionally, the Iberian–Ashkenazic DNA Project has a man named Delgado from Spain whose DNA markers are very close to those of Ballard at the 24-marker level. Thus, we believe that Ballard was likely Ashkenazi Jewish in ancestry.

William Ballard, b. 1617, d. 1689 Andover, MA R-L48, R-FT334262
13 23 14 10 11-14 12 12 11 14 13 30 19 9-10 11 11 23 15 19 29 15 16 17 19 10 19-22 17 15 17 36

### 4.8. Barney

The closest DNA haplotype matches we could find for the Barney lineage were in Bosnia, Bulgaria, and Serbia in the Balkan DNA Project, and the same held true in the I-p37 Haplogroup Project. These DNA data are not necessarily Jewish in ethnicity, but they do indicate that the Barney lineage is not Anglo-Saxon. Barney and Barnes are, however, Ashkenazic Jewish given names in the United States, as in the citations below:

Barnes-Jewish Hospital St. Louis
Barneys Clothing store in NYC owned by a man named Benjamin Ashkenazi
Camp Barney Mednitz (Jewish overnight camp, Atlanta, GA)

Jacob Barney, b. 1601 England
I-P37 23 16 10 12-15 11 15 12 14 11 18 8-9 11 11 26 14 18 29 11-14-14 -15 10 10 21-21

### 4.9. Bartlett

There are three different haplotypes available for Bartletts in England who immigrated to the New England Colony. Robert Bartlett's haplotype is in haplogroup G-m201, which is of Middle Eastern origin and one of the dominant haplogroups within the Jewish People (www.jewishgen.com, accessed multiple times from 8 January to 23 April 2021). However, we could find no close matches to this haplotype in any available database.

A second Bartlett lineage is in the I-m223 haplogroup. A search through several databases found no exact or close match to this haplotype either. However, there was a large set of haplotypes called "Dinaric-Unassigned" in the I-2a DNA Project that had several haplotypes that were only one or two markers off across the 24-marker set. Most of these men came from Belarus, Poland, Croatia, Greece, and Bulgaria. Notably the Dinaric ethnic label differentiates this group from the Anglo-Saxon Nordic racial category used by Madison Grant and other White Supremacists; this 'racial type' is described below:

The concept of a Dinaric race originated with Joseph Deniker in the late 19th century, but became most closely associated with the writings of Carleton Coon and Nazi eugenicist Hans Gunther. The term was derived from the Dinaric Alps in Southeastern Europe. The Dinaric type is said to be characterized by light skin, dark hair, a wide range of eye color; tall stature, a broad skull, long face, a narrow and prominent nose, sometimes aquiline; and a slender body type.

We have included some photos of Dinaric-type facial features in the Appendix A.

The third Bartlett lineage, I-L22, was a weak match (20 out of 24 markers) to three persons in the Iberian–Ashkenazi DNA Project whose surnames are Jordan, Garcia, and Wise. Thus, this lineage is likely of Ashkenazic Jewish descent.

| 220385 | Bartlett | Robert Bartlett, b. 1612and d.1675 | England | G-M201 | 15 23 15 10 15-15 11 12 12 12 10 29 17 9-9 11 12 25 16 21 28 13-13-13-14 |
|---|---|---|---|---|---|
| | Bartlett | Joseph Bartlett b. about 1630 and d. 1702 | England | I-M223 | 14 23 16 10 15-16 11 13 11 13 12 29 15 8-10 11 11 27 15 20 26 |
| Bartlett | Richard Bartlett b. 1575 d. 1647 | England | I-Y91123 | 13 | 23 15 10 14-14 11 14 11 12 11 18 28-9 12 12 23 35 99 16 10 19-21 14 14 46 1 36-37 20 18 15-15 8 10 89 12 23-25 |
| 41869 | Bartlett | Richard Bartlett | England | I-L22 | 12 13 25 35 10 14-14 14 11 12 18 15 8-9 12 23 59 29 12-14-15-16 |

### 4.10. Cabell

The two Capell/Cabell men from England were a 22 out of 24 match with a Jewish man from the Ayt Gennad tribe in Algeria who is in the Jewish R1b/R-m269 Project. Thus, this lineage is likely Jewish.

| 126477 | Capell | Edward Capell bc1683 Eydon, Northants, England | England | R-A10622 | 12 11 12- 1 1 1 1 1 2 1 9- 1 1 2 1 1 2 15-16- 1 1 19- 1 1 1 1 36- 1 1 1 16 1 1 1 1 1 23- 1 1 1 1 1 1 2 2 1 1 1 1 1 1 1 3 4 4 1 14 2 2 1 3 3 9 8 9 1 1 4 5 8 9 16-16 11 23 6 4 6 7 38 3 2 1 9-16 8 0 0 8 0 0 2 23 5 0 2 2 5 2 2 1 3 2 1 3 1 1 3 |
| IN95064 | Capell | Edward Capell 1657-1719 | England | R-M269 | 12 11 12- 1 1 1 1 1 2 1 9- 1 1 2 1 1 2 15-16- 1 1 19- 1 1 1 1 36- 1 1 1 16 1 1 3 4 4 1 14 2 2 1 3 3 9 8 9 1 1 4 5 9 9 16-16 11 23 7 4 6 7 38 3 2 1 9-16 8 0 0 |

### 4.11. Chapman

Chapman is a surname that is frequently Jewish, because it is an English term for merchant or vendor. The Edward Chapman lineage is E-m35, which, as already discussed, is Semitic and one of the founding lineages of the Jewish people. Within the Iberian–Ashkenazic Jewish Project, the Chapman haplotype is closest to persons from Ukraine, Poland, Romania, and Belarus. It is therefore likely that this family entered England with William I in 1066, while genetic "cousins" remained in France and later migrated to Eastern Europe.

| 95112 | Edward Chapman. 1612-1678 | England | E-BY6132 | 13 24 13 10 16-19 11 12 13 13 11 30 15 9-9 11 11 26 14 20 34 14-16-17-18 9 11 19-21 16 12 18 19 32-35 11 10 108 15-15 8 11 10 8 12 100 23-24 19 11 12 12 18 7 12 22 18 11 13 12 14 11 11 11 |

### 4.12. Church

The Church lineage belongs to a subgroup of I-p37 termed IL233. The closest haplotype we could find to this set of markers was a man named Smit from the Netherlands in the Iberian–Ashkenazi DNA Project who matched on 19 out of 24 markers. Thus, we cannot conclude that the Church lineage is Jewish.

| Church | Richard Church, ca1608-Unknown Dec 27 1668 | I-A19485 Origin | 13 23 15 10 14 12- 11 15 12 14 11 30 19 8- 11 11 25 14 17 29 11-14- 21- 34- 16- 18- 14-15 1010 21 14 10 16 17 34 13 10 118 17 8 11 10 8 12 1112 18 10 12 12 16 8 14 27 20 11 14 12 13 10 11 12 11 |

### 4.13. Cole

The Cole lineage is E-m35, which, as already noted, is Semitic and considered a founding ancestry of the Jewish people. There was a close match (22 out of 24 markers) with man in the Iberian–Ashkenazi Project named Edmund Freeman (1596–1682) who had also emigrated from England to Massachusetts with the Puritan settlers. Within the same phylogenetic cluster, there were persons named Castro, Cordoba, Lujan, Ha Levi, and Lucas (Germany). Thus, we conclude that the Cole lineage is Ashkenazi Jewish with ancestry in the Iberian Peninsula.

| 2 B542 Col 2 0 e | James Cole, b. 1600 and d. 1692 | E-m35 | 13 24 13 9 14-14 11 12 10 14 11 30 20 9-9 11 12 16 13 36-36 1 1 1 8 2 0 0 15-15 8 0 0 8 0 1 1 0 21-22 1 1 1 1 1 9 1 2 2 7 7 1 2 1 1 2 7 8 2 |

### 4.14. Conant

The Conant DNA markers shown below were a 22 out of 24 match to two persons in the Jewish R-1b/R-m269 project whose surnames are Rosales and Doran. Notably, the

Doran entry was from Ireland, indicating that some Jews made it to the Emerald Isle and kept knowledge of their Jewish ethnicity intact. It is therefore likely that this Conant line is Jewish.

| Conant | England | R-M269 | 13 | 25 14 10 11-14 12 12 11 13 13 29 17 9-10 11 11 25 |
|---|---|---|---|---|

### 4.15. Craft

The surname Craft/Kraft is a relatively common Ashkenazic Jewish surname. When the markers for this Massachusetts Bay settler were compared with current Jews listed in the Jewish R1b/R-m269 project database, several matches (21 out of 24 markers) were found. These matches came from persons originating in the Czech Republic, Austria, Germany, and Poland. Thus, the Craft family likely immigrated to England from Western Europe shortly after the arrival of William the Conqueror in 1066, while others sharing this DNA haplotype moved toward Eastern Europe as pogroms against Western European Jews increased in the Middle Ages.

| 126499 | Edward Craft, b. 30 Dec 1690 Boston, MA | England | R-M269 | 13 23 14 10 | 12-14 | 12 12 11 13 13 29 17 | 9-10 | 11 11 25 15 19 29 | 15-15-17-17 | 11 10 | 19-23 | 17 15 19 17 | 36-38 | 12 12 11 9 | 14-16 | 8 10 10 8 10 10 12 2 |
|---|---|---|---|---|---|---|---|---|---|---|---|---|---|---|---|---|

### 4.16. Crane

The Crane DNA markers shown below are very close to those found in an earlier study that included Confederate General Robert E. Lee. The Lee family immigrated to the Virginia Colony in the early 1600s from England. The Lee (and Crane) markers matched Jewish men in the Jewish Heritage DNA Project at a 22 out of 24 marker level. We conclude that the Crane Massachusetts settler likely is also of Ashkenazic Jewish ancestry.

| 134846 | Jasper Crane England I-m253 13 23 14 10 14-14 11 14 11 12 11 28 16 8-9 8 11 24 16 19 29 b. 1605 England, d. 1681 Newark, NJ |
|---|---|

### 4.17. Denney

Recall that the Denney/Dene surname was discussed in the Introduction as possibly being a Norman-French Ashkenazi surname. The Denney markers shown below are matches (20 out of 24 markers) to three men in the Jewish R1b/R-m269 project who come from Poland and Ukraine and one Jewish man (FitzAlan) who comes from Scotland (born 1267, died 1302). It is possible that FitzAlan was one of the Ashkenazi Jews brought to England in 1066 and that his family moved to Scotland when Jews were exiled from England in 1290. If this scenario is correct, then the Denney ancestor likely chose to remain in England as a crypto-Jew after 1290 and then immigrated to Massachusetts in the early 1600s with other English crypto-Jews.

| R-M269 | 13 | 24 | 1 1 11-4 2 14 | 1 1 1 1 1 3 2 2 0 4 3 0 | 1 9-6 10 | 1 1 2 1 1 2 1 1 4 5 9 9 | 15-15-17-17 | 1 1 0 1 | 19-23 | 1 1 1 1 6 5 8 8 | 38-38 | 1 1 2 2 1 | 9 | 15-16 | 8 0 0 | 1 8 0 0 | 1 1 1 0 0 2 | 23-23 | 1 1 1 1 6 0 2 2 | 1 8 4 | 1 2 2 1 1 1 1 1 1 1 2 2 0 4 2 2 3 1 1 4 2 |
|---|---|---|---|---|---|---|---|---|---|---|---|---|---|---|---|---|---|---|---|---|---|
| Den-ney | United King-dom | | | | | | | | | | | | | | | | | | | | |

### 4.18. Davenport

The Davenport DNA markers are shown below; they are close (20 out of 24 markers) matches with three men in the Jewish R1-b/R-m269 Project. Two of the men are from Lithuania and Poland, while the third, Fitzalan, was discussed with regard to the Denney lineage above. This repetitive pattern suggests that the Davenport, Denney, and Fitzalan ancestor(s) were likely Ashkenazic Jews who came from Normandy France to England around 1066 with William the Conqueror. While the Fitzalan genetic "cousin" may have moved over the border from England to Scotland in response to the expulsion of Jews from

England in 1290, both the Davenports and the Denneys may have become crypto-Jews and remained in England until they could make their way to North America in the early 1600s.

Richard Davenport 1606-1665 England, Scotland
13 24 15 10 11-14 12 12 12 13 13 29 18 8-10 11 11 25 15 19 28 15-15-17-18 11 11 19-13

### 4.19. Devereaux

The Deveraux DNA markers below are from a haplogroup, R-m512, known to be widespread among Ashkenazic Jewish Levites (www.jewishgen.com, accessed multiple times from 8 January to 23 April 2021). The name Deveraux is of French origin; thus, it is likely that the Deveraux Ashkenazic Jewish lineage arrived in England circa-1066 with William the Conqueror. The Deveraux family has a lengthy history among the nobility of England:

> William Devereux, Baron Devereux of Lyonshall (c. 1244–1314) was an English noble who was an important Marcher Lord as he held Lyonshall Castle controlling a strategically vital approach to the border of Wales in the time of Edward I and Edward II. He was the first of this family officially called to Parliament, and was ancestor to John Devereux, 1st Baron Devereux of Whitchurch Maund, the Devereux Earls of Essex, and the Devereux Viscounts of Hereford. Brook (2008)

> It is very significant that one of the Deveraux descendants would choose to come to the Massachusetts Bay Colony in the early 1600s, because it suggests that the family may have maintained its Jewish ancestry in secret for several centuries.

R-1a Devereux of Ireland and England: Lineage 2, Descendants of Jonathan Devereaux (Connecticut)

| | | | | |
|---|---|---|---|---|
| 76283 | England | R-M512 | 13 25 15 10 11-14 12 10 10 13 11 32 15 9-10 11 11 25 14 19 32 | 12-14-14-17 |
| 636663 | England | R-L2 | 13 24 13 10 11-14 12 12 11 13 13 29 16 9-10 11 11 25 15 19 29 | 15-16-17-17 11 11 19-23 16 15 17 17 34-3 |

### 4.20. Eaton

The Eaton DNA markers shown below form a cluster with several others discussed above in the Massachusetts Bay Colony, e.g., Davenport, Denney, and Fitzalan, belonging to haplogroup R-m269. This may indicate that these settlers shared a common ancestor at some point in time. Some descendants of the original ancestor seem to have migrated eastward to Poland and Lithuania and are Ashkenazic Jews in the Jewish R1b/R-m269 Project.

| | | | | | |
|---|---|---|---|---|---|
| 24 | 176290 | William Eaton, Dover, Kent England d. 1581 | England | R-M269 | 13 24 14 11 11-15 12 12 12 13 23 19 9-8 10 11 11 25 15 19 31 15-15-16-17 12 11 18-23 16 15 18 19 38-40 12 12 |
| 26 | 41600 | John Eaton, b.c.1636, Watertown, MA | England | R-M269 | 13 26 14 11 11-15 12 12 12 13 13 29 17 9-10 11 11 25 15 19 31 15-15-16-17 11 |

### 4.21. Gillett

The DNA markers for Gillett are close to those of settler Crane discussed earlier. They are in the I-L22 subgroup of the I-m253 haplogroup. Although I-m253 is primarily linked to Scandinavia, the L22 subclade evolved only about 3000 years ago. It is a northern European clade, but it is not as far north in origin as the Scandinavian haplogroup (L22 Haplogroup DNA Project website). In examining the website entries, several persons were identified as having likely Judaic roots; these included Maloret (France), Ortiz (Spain), Simonson

(Finland), Bromierz (Poland), Jacob Schneider (Hungary), and Micajah Brown (Appalachia). Thus, we believe this family is also of Jewish origin.

Jonathan Gillett
13 23 15 10 14-14 11 14 12 12
United States I-M253
11 28 16 8-9 8 11 23 16 20 28

### 4.22. Gibbs

The Gibbs DNA markers below are in the same R1-b/R-m269 Ashkenazi Jewish cluster found for Eaton, Denney, and Fitzalan, but are most harmonious with those of a man named Gottlieb Millke from Posen, Prussia (now Poland). Again, the deduction here is that these persons all shared a common Ashkenazi Jewish ancestor in Northern France at some time prior to 1066 and spread out after that—some descendants immigrating to Britain, while others remained in France and migrated eastward.

| 1 | Giles Gibbs (Somerset/ Dorset, ENG, c.1600) | R-M269 | 13 | 23 | 14 | 12 | 11-14 | 12 | 12 | 12 | 13 | 13 | 29 | 18 | 9-10 | 11 | 11 | 24 | 15 | 19 | 28 | 15-16-17-18 | 11 | 10 | 19-22 | 18 | 15 |
| 111998 | Jeremiah Gibbs b abt 1769 and d 1858 | R-M269 | 13 | 23 | 14 | 12 | 11-14 | 12 | 12 | 12 | 13 | 13 | 29 | 18 | 9-10 | 11 | 11 | 24 | 15 | 19 | 28 | 15-16-17-18 | 11 | 10 | 19-23 | 18 | 15 |

### 4.23. Goss

The Goss DNA haplotype shown below is from a haplogroup originating in Western Europe. A search through the I-m223 DNA Project revealed that the Goss haplotype was a close match to persons from Switzerland, Northern Ireland, Brazil and Germany. The surnames of these men included Fleming (i.e., from Flanders), Graf (Switzerland), Grein (1600, Germany, likely originally Gruen/Green), and Mathoso (Brazil). We cannot conclude that this Goss settler is Jewish, given this information. However, a search of the surname Goss brought up multiple examples of current Ashkenazic Jewish persons surnamed Goss. Some of these are excerpted below:

"Ken Goss brings to his (synagogue) presidency a unique background, a set of skills, humility, and a generous spirit that are certain to serve him well in the next two years. Having grown up in California and graduated from UCLA, he went on to pursue degrees in Jewish education and administration at the University of Judaism (today the American Jewish University). He became a teacher and an administrator in a Los Angeles Hebrew high school, a youth director, an educator and school principal in a synagogue, and then was named Assistant Director of Camp Ramah in California". (adatshalom.org/president-ken-goss, accessed multiple times from 8 January to 23 April 2021)

"Brandon Goss, a 28-year-old real estate agent who grew up in Northeast Ohio, will be one of 31 contestants vying for Clare Crawley's heart on the 16th season of "The Bachelorette". Goss graduated from Ohio State University in 2015, and was a member of Sigma Alpha Mu, a historically Jewish fraternity. He now lives in New York City and works for Corcoran, a real estate agency serving markets in New York City, The Hamptons, and South Florida." (bachelor-nation.fandom.com/wiki/Brandon_Goss, accessed multiple times from 8 January to 23 April 2021).

| Philip Goss b. c 1650 and d. 1696 | England | I-Y47901 | 15 | 23 | 16 10 115-15 11 13 12 14 12 32 15 8-10 11 11 | 10 | 15-15 | 11 | 15 | 11 | 12 |

### 4.24. Hardy

The Hardy DNA scores shown below were quite similar (22 out of 24 markers) to two men of Portuguese ancestry in the Iberian–Ashkenazi Jewish DNA Project; the men were named Julio Pereira and Sar Shushan. Thus, it is likely that Hardy is of Ashkenazic Jewish ancestry. Notably, the surname Hardy is archaic French for "strong"; this suggests that the original ancestor may have emigrated from France to England during the reign of William I.

| 43 | 113330 | Hardy | John (son of Thomas) Hardy, b.c.1646, Ipswich, MA | R-U106 | 13 | 24 | 14 | 10 | 11-14 | 12 | 12 | 12 | 13 | 13 | 29 | 17 | 9-10 | 11 | 11 | 25 | 15 | 19 | 30 | -18 |

### 4.25. Ludlow

The Ludlow markers were very similar to two groups of Jewish persons in different FTDNA projects. The first cluster is from the Jewish R1b/R-m269 Project; these were a man named Stein from Austria, a man named Lewinski from Poland, and an unnamed Jewish man from Russia. In the Iberian–Ashkenazic Jewish Project, the Ludlow markers were very close to those of an Espinoza and Latre from Spain. This suggests that the Ludlow ancestor was probably Ashkenazic Jewish and that genetic "cousins" migrated to Spain and Eastern Europe.

| 7862 | LUDLOW | Stephen Ludlow b. about 1500 | England | R-M269 | 13 | 24 | 14 | 11 | 11-14 | 12 | 12 | 11 | 13 | 13 | 29 |

### 4.26. Parker

Before turning to these DNA results, we must note that one of the names in the set below is Elihu Parker; Elihu is Hebrew for "Jehovah is God" and is often used as a given name among Jewish people. The DNA markers for the New England Parker family are a close match (21 out of 24 markers) to three men in the Jewish R1b/R-m269 Project who come from Belarus, Poland, and Hungary. Thus, we believe that the Parker ancestor was likely an Ashkenazic Jew.

Elihu Parker, Thomas Parker, Jonathan Parker, John Parker, England, Connecticut, Rhode Island, Vermont, NY R-BY39898, 1609,1683
13 23 14 10 11-14 12 12 12 13 13 29 18 9-10 11 11 25 14 19 30 8 10 10 8 10 11 12 23-23

### 4.27. Pratt

The Pratt DNA markers shown below were closely matched to multiple members of the Iberian–Ashkenazic Project. These included two men with Spanish surnames from Spain and Portugal; but there was also a close match to a man named Stone of unknown origin. Stone may have been Stein at an earlier date. A slightly more distant relationship was found to a man named Thomas Alligood from Jamestown, Virginia [1627], also in the Iberian–Ashkenazic Project. Thus, it is very likely that the Pratt family was of Ashkenazic Jewish origin.

| 41630 | Pratt | England | R-M269 | 13 | 24 | 14 | 11 | 11-14 | 12 | 12 | 12 | 13 | 13 | 30 | 18 | 9-10 | 11 | 11 | 26 | 15 | 19 | 29 |

### 4.28. Revell

The Revell DNA markers below show a match to a group of Ashkenazi Jews from Eastern Europe, e.g., Russian Federation, Ukraine, Poland, at the 20 out of 24 marker level. Notably, there is a member of the Revell group below who comes from France, and there is also a close match to the Scottish man surnamed Fitzalan, mentioned earlier. Thus, the Revells also belong to the R1b/R-m269 cluster, e.g., Davenport and Denney, discussed earlier.

Revell England, France R-m269
13 23 14 11 11-14 12 12 11 14 13 30 17 9-10 11 11 24 15 19 29 15-16-17-18 11 10 19-23

### 4.29. Sargeant

The Sargeant DNA markers come from Haplogroup E-m35; as discussed earlier, this is a Semitic-originating haplogroup and one of the founding lineages of the Jewish people. These markers are close (21 out of 24 markers) to two men in the Iberian–Ashkenazi Project named Mizruchi and Greengard from Ukraine and Lithuania. Thus, it is likely that this Sargeant had Jewish ancestry.

William Sargent (b. 1603, Bath, England; d.1675 -Amesbury, MA)
E-m35 13 24 13 10 16-18 11 12 12 13 11 30 15 9-9 11 11 27 14 20 33 14-15-16-17 9 10 19-21 16 12 19 31-35 11 10 10 8 15-15 8 11 10 12 10

*4.30. Sprague*

The Sprague DNA markers shown below were a close match (21 out of 24 markers) to a man named Benjamin Raphael in the Family Tree DNA Jewish R1b/R-m269 project. The match person was from Germany, indicating that he was likely Ashkenazic. Thus, the Sprague ancestry is also likely Ashkenazic Jewish. (Sprague may be derived from sprechen—to speak—as in "Sprechen Sie Deutsch?".

Sprague, Edward, b. 1576, d. 1614 England
Sprague, William, b. 1609, Upwey, England
Sprague, Thomas b.1804, Marion County, Ohio
R-m269 14 23 14 11 11-14 12 12 11 13 13 29 16 9-10 11 11 24 15 19 29 15-16-18-18 11 10 19-23 17 15 17 17

*4.31. Stearns*

Stern is a common Ashkenazic Jewish surname derived from the Germanic word stern/star. In the Appalachian Region, this surname became Starnes and is of Jewish ancestry (Hirschman et al. 2019a). This Stearns haplotype is close to that of John William Bell in the Iberian–Ashkenazi DNA Project—22 out of 24 markers. It is likely that this Massachusetts Bay Colony settler has Ashkenazic Jewish ancestry.

| | Edgar Stearns | | England | G-M201 | 14 | 22 | 15 | 10 | 13-15 | 11 | 12 | 12 | 12 | 11 | 29 | |
|---|---|---|---|---|---|---|---|---|---|---|---|---|---|---|---|---|
| | Leonard Starns, b.c, 1726, German Flatts, NY G-M201 15 22 15 10 13-15 11 12 13 12 11 29 | | | | | | | | | | | | 29 | | | |

*4.32. Stone*

This is a common Ashkenazic surname derived from the Germanic Stein. The G-L13 subclade is most common in north central Europe. The closest match we could find for this haplotype is a man surnamed Silva who lived in New Mexico and is in the Iberian–Ashkenazi Project. The match was at the 19 out of 24 level.

Hugh Stone, Joseph Stone, Samuel Stone

G-L13, G-Z42474 14 22 15 10 12-14 11 12 13 12 11 29 20 9-9 11 11 23 16 21 27 8 11 10 83 11 13 21-22 14 10

*4.33. Sweet*

The surname Sweet is common among Ashkenazi Jews as Suss, Suess, and Sussman (Germanic for Sweet); Jews were early merchants of sugar, chocolate, and candies (Golb 1998). The DNA haplotype markers below are close to those of an Iberian–Ashkenazic Project member named Dourado from Portugal and another man named Stone. Thus, it is likely that Massachusetts Bay colonist Sweet was of Ashkenazic Jewish descent.

Sweet

| R-M269 | 13 24 14 10 | 11-14 12 12 | 11 | 14 13 30 18 | 9-10 11 11 | 25 15 19 29 | 15-15-17-17 | 11 10 | 19-23 | 15 15 17 18 | 38-38 | 12 12 |
|---|---|---|---|---|---|---|---|---|---|---|---|---|
| R-M269 | 13 24 14 10 | 11-14 12 12 | 11 | 14 13 30 18 | 9-10 11 11 | 25 15 19 29 | 15-15-17-18 | 11 10 | 19-23 | 15 15 17 18 | 36-38 | 12 12 |

*4.34. Sylvester*

The surname Sylvester—and its translations: Bloise, Woods, and Forest—is common among Sephardic Jews (see Sephardim.co, date accesed). There was a match (20 out of 24 markers) with Tzvi Persky from Belarus in the Jewish R1b/Rm269 project and also similar matches to several Iberian–Ashkenazi Project members who reside in Spain, Portugal, and Scotland.

Richard Sylvester, b. ca. 1600 England, Nathan Brown Sylvester, b. 1821 U.S.

R-M269 13 24 14 11 11-14 12 12 12 12 14 27 19 9-10 11 11 26 15 19 29 15-15-17-18 10 12 19-23 16 15 17 18

*4.35. Taylor*

The Taylor DNA markers are close (20 out of 24) to those of John William Bell of England who is a member of the Iberian–Ashkenazi DNA Project. Thus, it is likely that Taylor was also of Ashkenazic Jewish ancestry and that he and John William Bell share a common ancestor. Additionally, the G-m201 haplogroup is one of the primary ancestries of the Jewish people (www.jewishgen.com, accessed multiple times from 8 January to 23 April 2021).

| | | | | |
|---|---|---|---|---|
| Taylor | Edward Taylor, b. 1649 and d. 1710 | England | G-M201 | 15 21 15 10 13-15 10 12 13 12 11 28 15 9-9 11 11 23 16 21 30 12-12-12-12 10 10 19-19 15 12 15 18 33-33 11 10 |
| Taylor | | Unknown Origin | G-M201 | 15 21 15 10 14-15 10 12 13 12 11 28 15 9-9 11 11 23 16 22 30 12-12-12-12 10 10 19-19 15 12 15 18 34-34 11 10 11 8 15-17 8 11 10 8 11 10 14 22-22 17 11 12 12 15 8 14 22 19 15 13 11 13 10 11 |
| Taylor | Edward Taylor. d. 1710 NJ | Unknown Origin | G-M3302 | 16 21 15 10 14-15 10 12 13 12 11 28 15 9-9 11 11 23 16 22 30 12-12-12-12 10 10 19-19 15 12 15 18 33-33 11 10 11 8 15-16 8 11 10 8 11 10 14 22-22 17 10 12 12 15 8 13 22 19 15 13 11 13 10 11 |

*4.36. Tollemache/Talmadge*

The Tollemache/Talmadge DNA markers are a close match (20 out of 24 markers) with several persons in the Balkan Genetics DNA Project and also the Jewish Heritage Project. These include persons from Poland, Russian Federation, Bulgaria, Croatia, and a William Crabtree from England. The Crabtree surname is found among the Jewish settlers in Central Appalachia (see Hirschman et al. 2019a). Thus, we believe that it is likely the ancestor was of Ashkenazic Jewish ancestry.

The Tollemache surname has a fascinating history: Recorded in many forms including Talmasche, Talmadge, Talmage, Tallamach, Tolemarche, Tolemache, Tollemache, and Tollmache, this very interesting and unusual surname . . . is occupational and of pre-10th century Old French origins. It derives from the word "talemache", meaning "one who carries a knap-sack", and therefore . . . it is possible that it could refer to a messenger or even a merchant. The word was introduced into England after the famous Conquest of 1066. Early examples of the surname recording include Hugo Talamasche in the Pipe Rolls of the county of Oxford in the year 1130, Robert Talemasche also of Oxfordshire in the Eynsham charters dated 1150, and William Talemach in the Ministers Accounts of the Earldom of Cornwall in the year 1297 (https://www.surnamedb.com/Surname/Tollemache#ixzz6nxh1N0Z9, accessed multiple times from 8 January to 23 April 2021).

Thus, together with the Devereaux surname discussed earlier, we now have two instances of ennobled families immigrating to the Massachusetts Bay Colony from England. One of the questions this raises is did these crypto-Jewish nobles communicate with one another across the time lapse of three centuries and with the other settlers who ventured to New England with them?

In one of the great twists of ethnic history, a twentieth century descendant of the Tollemache family, Eugene Talmadge, became Governor of Georgia three times and was noted for his racist White Supremacist views:

"Eugene Talmadge, served three terms as an avowedly racist governor and went to his maker wholly unreconciled to the outcome of the civil war. His son, Herman, was cast in the same mould and, on Eugene's death, took control of a relentless local party machine as vicious as those in New York and Chicago. It shamelessly espoused redneck policies and Talmadge greeted the 1954 Supreme Court ban on school segregation with the ominous forecast that "blood will run in the streets of Atlanta". (The Guardian, Sun 24 Mar 2002 20.25 EST).

22083TollemacheEnglandI-M2531322151013-14111412131129168-98112316203013-14-15-16101019-211414171933-35

## 5. Limitations and Future Research

This research has examined male colonists emigrating from England to the Massachusetts Bay Colony in the early 1600s for whom DNA samples are available. By using public genealogical databases, we have shown that most were likely of Ashkenazic Jewish descent. By linking DNA from several of the settlers to Ashkenazic Jewish men in Russia, Poland, Belarus, Latvia, and other places in Eastern Europe, we have supported the hypothesis that these same genes are also found in the Jewish "morass of human misery" decried by Madison Grant in his White Protestant Supremacist writings during the 1930s and echoed today by various far-right groups. Reminiscent of the Pogo cartoon slogan, "we have met the enemy and he is us," we must now grasp the irony that WASP Colonial Massachusetts likely never existed.

The present research results, while supportive of our thesis, are weakened by the lack of MtDNA results. It would be of great benefit if persons having strong genealogical documentation of their maternal lineage dating back to the settling of Massachusetts would contribute their MtDNA results to a public database so that researchers could verify or disconfirm the findings reported here for male colonial settlers. If future MtDNA examination confirms Jewish ancestry for the women of early Massachusetts, then the likelihood that these colonists maintained knowledge of their ethnic identity would be greatly strength-

ened. Perhaps diaries, letters, grave markers, or memories of great grandparents' behaviors would reveal that these settlers maintained some Judaic practices, just as has been found for many Hispanic families in the American southwest (e.g., Kagan and Dyer 2011).

**Funding:** This research received no external funding.

**Institutional Review Board Statement:** This research received no external funding.

**Data Availability Statement:** Not applicable.

**Conflicts of Interest:** The author declares no conflict of interest.

## Appendix A. Images of the Puritan Settlers, Persons of Dinaric Ancestry and Semitic Ancestry

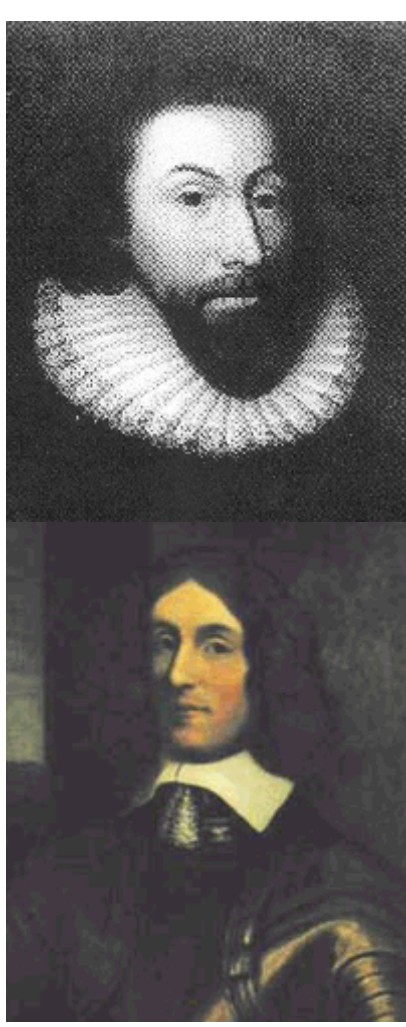
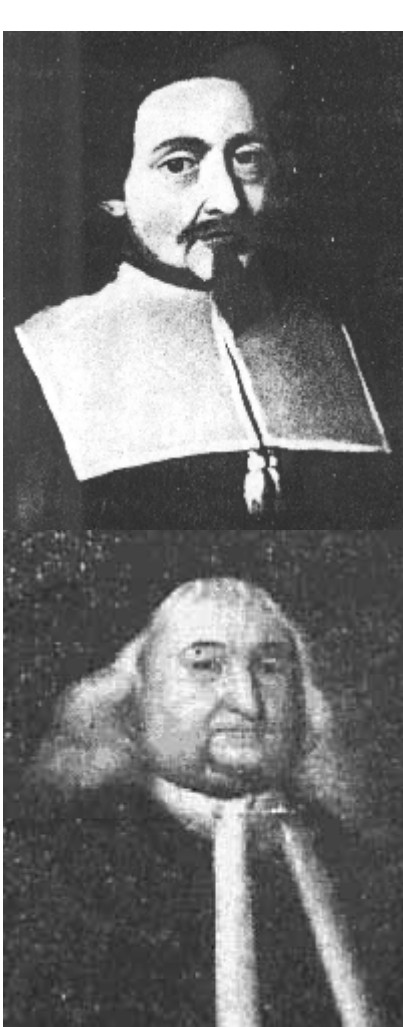

**Figure A1.** *Cont.*

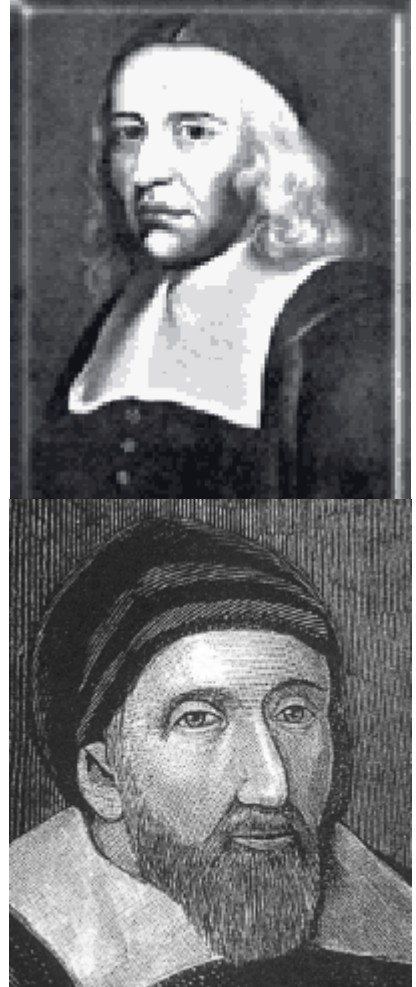
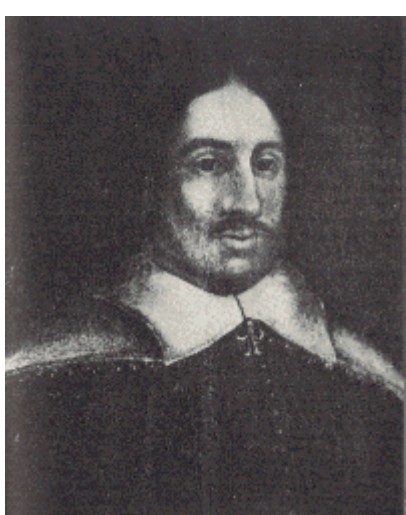
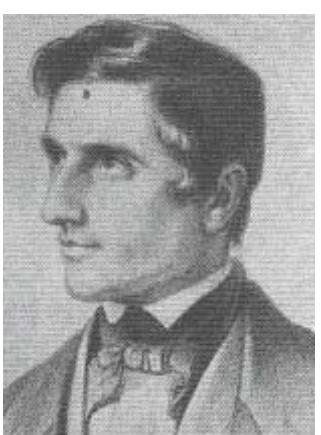

**Figure A1.** Puritan Settlers (Taken from the Winthrop Society website).

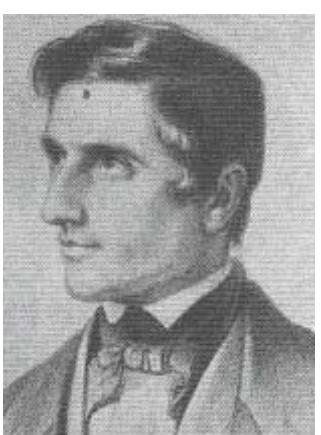
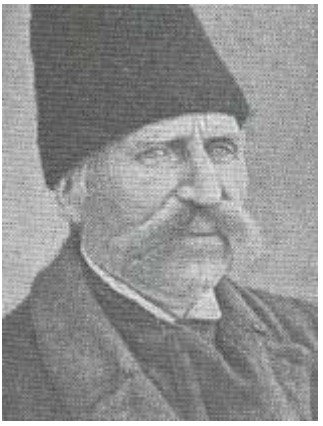
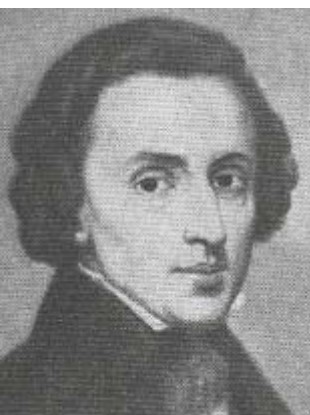

**Figure A2.** Persons of Dinaric Ancestry.

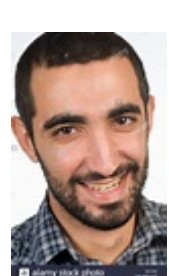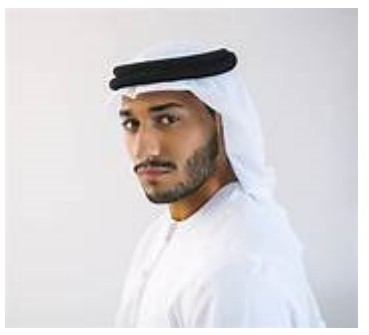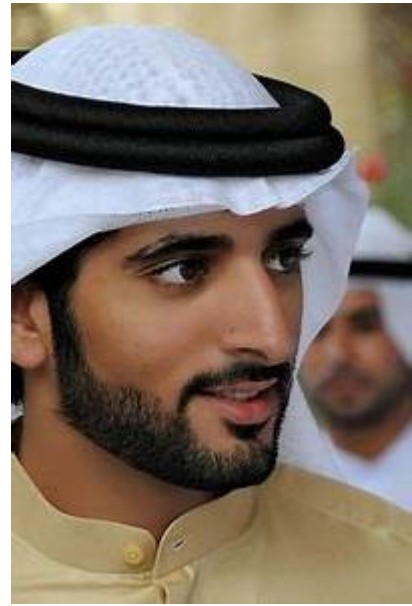

**Figure A3.** Persons of Middle Eastern Ancestry.

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
