# Peer review of "Challenging the Fundamental Premise of White Supremacy: DNA Documents the Jewish Origins of the New England Colony"

_socsci, doi:10.3390/socsci10060232_

Round 1

Reviewer 1 Report

This is an interesting article, and I am glad that I had the opportunity to review it.

I think that this is a very promising study with much potential.  I completely support the author’s argument that DNA analysis undermines many of the traditional assumptions about who we are, both as individuals and collectively as families, as nations, and as ethnic groups.  I also agree that perhaps the ultimate lesson of DNA supports an argument for tolerance and togetherness as it emphasizes the ties that bind humans rather than the issues that divide them.  Certainly DNA seems to undermine claims of any particular group for physical or intellectual superiority.

I do, however, have several questions and concerns and some possible suggestions regarding this essay.  The author, or authors, do convince me that some of the early New England settlers probably did have remote paternal Jewish ancestry.  As someone familiar with Y-DNA analysis and migration studies, this did not come as a particular surprise. 

Still, there seem to be a lot of gaps in this analysis – some that may never be able to be filled and others that might be filled but only after a great deal of additional DNA analysis and research of other types, including folkways and material culture.

The author is right to question the foundational myth associated with the Puritan “City on the Hill” and the idea, not just of white superiority, but of American exceptionalism that has stemmed from it.  Most serious historians recognize this as a false construct but one that has been highly influential in creating the narrative of the American past that has been told and retold for centuries.

In terms of the larger argument of this paper, I really question the assertion that the patterns revealed here amount to a crypto-Jewish community (and by crypto-Jew I mean a secret adherent of Judaism who outwardly professed another faith) that maintained a sense of origins, identity, and culture for 5 ½ centuries between their arrival in England following 1066 and their transplantation to New England between about 1620 and 1650.  There is nothing in the article that convinces me of that. 

It is interesting that a few families that seem to have shared Jewish Y-DNA all ended up in New England and intermarried there, but there are reasons other than a shared Jewish heritage centuries earlier that could account for that.  Had these families come to England following the Norman Conquest still as practicing Jews, they might have been acculturated into English society over the next five centuries and perhaps still continued to associate because of local ties in the areas from which they came.  Often villages and entire kinship networks relocated to New England; family ties were an important part of this migration.  Thus active Jewish identity (crypto Judaism) need not be an explanatory factor for why families with Jewish Y-DNA might have all ended up in New England.

One thing that this study lacks is a detailed examination of the individuals who headed the families discussed in this paper.  Who were they in England?  What was their occupation?  Did they own land?  Did they hold political office?  Were they united by ties of commerce, marriage, political affiliation, church participation?  Much of this work has already been done by Robert Charles Anderson, who has developed a very detailed study of the early New England migrants and their immediate families.  His studies fill dozens of volumes and trace some early colonists in the most minute detail, while others still remain obscure despite decades of research. 

The second thing that I think would strongly benefit this study would be cultural analysis along the lines of what David Hackett Fischer attempted in Albion’s Seed.  Fischer’s argument in many ways opposes the current author (or authors).  Fischer argues that New England cultural folkways – thinking about everything from naming patterns and food preparation to larger theological issues – was shaped by the cultural dynamics of the East Anglian villages from which most of these early colonists came, and that the roots of those folkways were deeply embedded in East Anglian society.  Some of these Fischer asserts may even go back to events before the Conquest.

However, Fischer’s analysis might not entirely differ with the current author. What would be really interesting would be if the author could find evidence of Jewish folkways embedded within the English village societies from which the New England migrants came.  If the current author is correct, and if this Jewish identity was strong enough to persist from 1066 to the 1630s, it must have left its imprint in other ways than just Y-DNA.  And if it did survive for centuries in England in these families, it seems very odd that that identity and culture would have disappeared entirely in America.

I question less the possible Jewish genetic origins of these families than the idea that these families were crypto-Jews at the time of they came to New England.  If they were, why they would come to the incredibly repressive New England society in the first place would be a problem that would need to be explained.  Any number of other possible destinations would seem to be more appealing than Puritan New England and its frigid geography.

If any early New England settlers were crypto-Jews who identified as such rather than English families of possible earlier Jewish lineage, another issue that should be considered is how they managed to conceal their Jewish identities in New England.  This was an incredibly repressive society in which sinfulness and difference were deliberately sought out.  The history of criminality in New England reveals just how closely individuals paid attention to what their neighbors were doing.  As examples, one might consult any one of dozens of scholarly community studies, but two of the best are John Demos’ Entertaining Satan:  Witchcraft and Culture in Early New England and Boyer and Nissenbaum’s Salem Possessed:  The Social Origins of Witchcraft.  Anything that was “different” was considered deviant and liable to accusation and punishment.

Were I writing this article myself, my technique would be to focus the first part of the article on the DNA evidence, and provide a strong case to argue that at least a few dozen of these early New England settlers may have been of Jewish Y-DNA origin.  I would then turn to a discussion of what this means.  The idea of remote Jewish origins could be discussed as well as the possibility of more recent crypto-Jewish origins in some families.  (Certainly there were Jews and crypto-Jews in England in the early 1600s.)  Then, if I chose to engage in an argument undermining assertions of “White Superiority,” I would tease this out in the concluding section rather than making it the central thrust of this article.  As it is currently structured, the article seems geared towards contemporary cultural and political concerns that will date it and possibly undermine its long term scholarly credibility.  A subtler approach might make a more nuanced case that lays the foundation for sustained research in this area for a long time to come.  Essentially, in my view, framing the piece almost in reverse would be a better technique.

While engaging with Grant makes an interesting framing device, my sense of Madison Grant is that his views were largely perceived as outdated by the time of his death in 1937 and shortly thereafter.  Some current extremist groups may take an interest in his ideas, but I am not aware of any serious scholar who today takes Grant’s ideas seriously.  Usually Grant is used as an example of the racist thinking that underlay many of the assumptions of eugenics and Nativism during the Progressive era – although the author is correct that his Nordic theory did have a greater impact in Nazi Germany.

Here are some specific comments and suggestions:

Lines 29-32: 

 But what if this image (Figure 1) is a fiction – an amalgam of racial ideologies grounded the false belief that North America was colonized by British Protestants of Anglo-Saxon ancestry. What if, in fact, the settlers of New England were not who we thought they were.

In one sense this is meaningful, but this interpretation discounts the nature vs. nurture debate.  Whatever the genetic origins of the Puritans (or any other group), their identity was largely a product of the larger culture in which they were reared – the values and folkways of their society.  If the author is correct that the early Puritan males had Ashkenazic Jewish ancestry, does that really change anything?  By the author’s own admission, they would have lived in England for nearly six centuries before coming to North America.  Moreover, Y-DNA is not all DNA.  The lone Y-chromosome mutates slowly across generations, but autosomal DNA, Mitochondrial DNA, and X-DNA reveal very different stories.  I might carry Y-DNA that indicates an ultimate North African paternal heritage, but 100% of my autosomal DNA might indicate non-African ancestry, as might my autosomal DNA and X-DNA.  So, first, let us suppose that the author is correct:  what does this really mean or change?

Is there anything in autosomal DNA to indicate Jewish heritage?  Because of centuries of intermarriage within fairly small groups, Jewish autosomal DNA often makes relationships appear closer than they are – for instance, people who are third cousins might share more autosomal DNA than other sets of third cousins because their ancestors were themselves related in some way, too. 

If these New England pioneers had Ashkenazic Jewish heritage and continued to intermarry as suggested in the article, there would likely be some autosomal evidence of Jewish heritage among older descendants living today, say people born in the 1930s.

Lines 32-34:

 What if they were actually Ashkenazic (Eastern European) Jews who were not only not Protestant, but not even ethnically British, and not even “white” as that term is usually applied? What happens then to the doctrine of White Supremacy?

Again, for the sake of argument, would having Ashkenazic Y-DNA mean an individual was “not even ‘white’ as that term is usually applied?”  Again, I would think autosomal DNA, X-DNA, and MT DNA would all be worth considering here to support this argument.

In a different vein, however, let us ask whether this “discovery” of Ashkenazic Jewish ancestry would be new and surprising?  The Biblical creation story places the origins of human life with the Biblical Adam and the Biblical Eve, who were fourteen generations removed Abraham, who in turn was fourteen generations removed from David.  “Father Abraham” was the ancestor of the twelve tribes of Israel.  Various theories have been purported for centuries to explain how the twelve tribes spread throughout the world to account for all human life. 

Would the Puritan immigrants have doubted the Biblical interpretation linking them in some way to Abraham, Isaac, Jacob, and the twelve tribes of Israel?  David Starr Jordan, writing in the 1930s, traced the lineage of the English and Scottish monarchs to Tephi, wife of Heremon and daughter of Zedekiah, who was supposedly the mother of Irial Faidh and herself fiftieth in descend from Adam and Eve and thirtieth in descent from Abraham and Sarah.  While they may have been loathe to identify with Jewish or Middle Eastern culture, individuals like this were not hesitant to make an ancient claim to Jewish ancestry.  (Hence the idea that the Blarney Stone might actually be the pillow upon which Jacob had his dream.)

The question would seem more to be a case of when the Jewish heritage and identity was “lost” – prior to 1066?  Between 1066 and 1300?  1066 and 1600?  1066 and 1800?  My own analysis would suggest an early date, probably prior to 1500 and possibly much earlier.

Lines 61-62

White Supremacy is – at its core – a theory of genetic ethnic differentiation. One’s genes carry one’s ethnicity, not one’s language or country of origin.

The quote apparently supporting this is from 1927.  Is there another source?  It needs to be indicated, and supported by something more recent, if this is used as the grounding for a whole system of thought.  As noted above, I’d suggest teasing this argument out more in the conclusion than making it the foundational framing mechanism for the entire essay.

Lines 84-85

The research we report here shows that this foundational assumption – that New England was settled by Anglo-Saxon/Nordic persons -- is grossly false. If this core tenet falls, then White Protestant Supremacy has no basis in fact or history.

Who argues that is a foundational assumption?  No serious historian today should hold these views, although the ones cited from the 1920s and 1930s (who were not leading academic historians of the time) might.

Even if you are correct, however, I don’t think this amounts to the foundational assumption being “grossly false.”  At most, you cover a few dozen families here, while estimates are that about 100,000 people came to New England before 1650.

Lines 106-107

Where is evidence to link this Denny/Dene family with the Hebrew Daniel?

Lines 143-147

While these families may (or may not) have had genetic Jewish Y-DNA ancestry, where is the evidence that they were crypto-Jews (i.e., retained Jewish faith and identity) across generations?

Lines 148-188

I agree with the premise here; analysis of this type has great potential value.

Lines 220-230

Heritage societies of this sort are a good example of misguided history and genealogy – again maybe not the best framing device.

Line 225

This should be a complete sentence:

(excerpt from the Winthrop Society Charter).

Lines 241-249

It would be better to cite a source from the website than the website itself.  This might be the journal itself.  The website URL may change and future readers might not know exactly what was being referenced.

260-265

This is interesting, but even if the article’s premise is correct, what do the references to Grant (a very obscure figure today) prove?  Moreover, there’s a lot of unpacking to do in this short paragraph…  White Superiority, Third Reich, Grant…

271-275

Although greatly advanced when compared with a decade ago, DNA analysis of genetic origins is still a new and developing field, and testing companies are continually refining their ethnic identifications.  I would urge caution when ascribing too much significance to the ethnic origins of any one family (Y-DNA haplotype) or individual (autosomal analysis and ethnicity typing).  Will these identifications hold up in 50 years?

Page 9, paragraph 2, immediately below Figure 2:

Is this really what the tree suggests?  It’s a big leap from 800-500 BCE to the unspecified arrival of the Abbotts in England.

Page 10, lines 3-10: 

Interesting, but 22/24 markers is hardly conclusive.  If a connection exists, it would have to be in the distant past.

Page 10, lines 17-24

Again, interesting:  But is it not possible that some of these families were the product of individuals who left England or northwestern Europe and left illegitimate offspring in these regions as a result of trace (Silk Road, Spice Trade) or conquest (Crusades, etc.)

This also leaves out the possibility of adoption and illegitimacy, as well as the reality that many of those taking the Y-DNA test assert (rather than prove) a particular ancestor.  For instance, if my name is Smith, I may list my ancestor as Sir John Smith, 17th Lord of Smitherton, because he is the most prominent person of that surname and the one most often claimed as ancestor.  It may be, however, that my ancestor was John Smith, chicken tender, who was of no relation, but some genealogist (as was common in the early 20th century genealogies, when aristocratic ancestors were what one desired) falsely linked the two.  So, my Y-DNA haplotype may reflect John Smith, chicken tender, even though a dozen of his descendants all list Sir John Smith as their ancestor. 

Page 10, lines 27-29

Interesting, but a big presumption

Page 11, lines 40-41

this Adams family is also very likely of Jewish ancestry

This leave a lot unexplained – no evidence of ancestry or earlier history prior to birth of John Adams about 1600.

Page 11, lines 49-51

20/24 is not a particularly close match, and many other scenarios could explain a possible shared lineage

Page 12, lines 58-63

The same concerns apply here

Pages 12-13, lines 85-89

20th century examples from the USA should not be used as examples of possible Jewish origins of an English family many centuries earlier

Page 13, lines 99-100

This should be databases, not data bases

Page 13, lines 114-115

Again, same concerns as above apply – this is a very big leap – (Jordan, Garcia, Wise, and Bartlett)

Page 15, lines 143-148

There was a close match (22 out of 24 markers) with man in the Iberian Ashkenazi Project named Edmund Freeman (1596 – 1682) who had also emigrated from England to Massachusetts with the Puritan settlers.

The match would actually be with a man who claims descend from Edmund Freeman, not with Edmund Freeman himself (see comment above; claims descent from – the lineage would also have to be documented to verify that this is the Edmund Freeman family Y-DNA).  Many people incorrectly claim descent from a particular individual in FTDNA identifications.  To be certain, the lineage would have to be documented in each generation and the match confirmed with multiple individuals bearing the surname who descend from this particular individual.

Page 16, lines 170-175

one Jewish man (FitzAlan) who comes from Scotland (born 1267, died 1302). It is possible that FitzAlan was one of the Ashkenazi Jews brought to England in 1066 and that his family moved to Scotland when Jews were exiled from England in 1290. If this scenario is correct, then the Denney ancestor likely chose to remain in England as a crypto-Jew after 1290 and then immigrated to Massachusetts in the early 1600s with other English crypto-Jews.

Again, unless this man’s body and DNA have been preserved, this is not a match to him but to an individual claiming descent from him – which may, or may not, be the case

Page 17, lines 200-203

Citation needs to be modified to fit journal style

Page 17, lines 203-205

It is very significant that one of the Deveraux descendants would choose to come to the Massachusetts Bay Colony in the early 1600s, because it suggests that the family maintained its Jewish ancestry in secret for several centuries.

Is this really what it suggests?  Again, too much significance is ascribed to these assertions with no corroborating evidence.  There were powerful historical reasons embedded within 17th century England for the “Great Migration” that might have drawn both families

Page 17, 219-220

What is Micajah Brown’s background?  Was he Melungeon?

Page 18, 237-250

Quotations should be in quotation marks; what are the sources for this information on Ken Goss and Brandon Goss?

Rev. Jesse Davis Goss (1793-1866) and Rev. Benajah Goss (1809-1866) were Baptist ministers in Georgia and Alabama; could we use this as evidence to make a conclusion about an early Puritan immigrant?

Page 18, 254

Omitted word:  Hardy was of

Page 19, 269-270

Elihu is used as a name in many different Judeo-Christian groups; I count approximately 75 men in the 1850 Georgia census named Elihu who belonged to Protestant families.  Elihu Yale of Massachusetts, whose family was from Cheshire, England, and North Wales, is another example (unless he is included in your database of Jewish Y-DNA families)

Page 23, lines 401-403

By using public genealogical databases, we have shown that virtually all were likely of Ashkenazic Jewish descent. These

Keep in mind that your sample is very small.  Tens of thousands of people came to New England during the Great Migration between 1620 and 1640. 

Robert Charles Anderson has done a fine job documenting the lives and backgrounds of early Puritan immigrants:

https://www.americanancestors.org/browse/publications/ongoing-study-projects/the-great-migration-study-project

Page 23, 444-448

Why would Protestant Puritanism – one of the most repressive systems of religious beliefs – be a “more comfortable fit” ?

Page 23, 446-450

If this were true – namely that these individuals preserved Jewish identity across six centuries (1066-1600s) – one would expect some archeological evidence as well as anthropological and oral history evidence.  Did these individuals cling their Jewish heritage in England for 600 years only to lose it in less than four centuries in North America?  The crypto Jews of northern Mexico and the Spanish borderlands preserved their Jewish heritage in oral tales as well as in religious artifacts, and archaeological excavations have supported this.  Even today, one meets families – Catholic and Spanish speaking – who will immediately explain that their earliest ancestors were Jews who came to New Spain. 

This article combines DNA evidence with other forms of cultural evidence to make a strong case for Jewish heritage in New Spain:  https://www.smithsonianmag.com/science-nature/the-secret-jews-of-san-luis-valley-11765512/

The best documented source for information on early New England settlers is the works of Robert Charles Anderson:

https://www.americanancestors.org/browse/publications/ongoing-study-projects/the-great-migration-study-project

Caution:  I do not regard The Winthrop Society as a very credible source and would urge caution in using any of their materials.  They appear to be purely interested in a pseudo-verified lineage to an early ancestor rather than in any serious research into the early New England colonists, their background, and their culture.

Reviewer 2 Report

I suggest a more sound historical source than www.avotaynu.com. A sentence like "During the 1200s anti-Semitic pogroms 239 forced most of the Jews in these Western European communities to begin migrating east- 240 ward" (lines 239-240), is not serious. The main reason for migration was economic. Life in Germany became more and more difficult for the Jews, but "pogroms" were not during the 13th century except for The Rintfleisch or Rindfleisch massacres against Jews in the year 1298. The migration to the east started before.

Further, in lines 243-246 the author quotes again from www.avotaynu.com regarding the Yiddish as the spoken language of the Ashkenazi Jews. Again, with all respect to this site, we don't know when and where was Yiddish developed. It was certainly not spoken in England, and therefore not relevant to the paper.

Round 2

Reviewer 1 Report

Response to Author’s Response:

Thank you for the interesting and detailed responses you provided.

Your concern about the public reaction to some of this information is understandable; there is also evidence that many southern white families had mixed race African and Native American ancestry during the 17th century (which could also include elements of Jewish heritage, as some of these families, like the Driggus/Driggers/Rodriguez families fall in to the category Ira Berlin calls “Atlantic Creoles” – individuals with varied heritage in many nations who were truly part of an “Atlantic world”).  This includes a number of notable individuals, similar to the Talmadge example you cite.  It is clear that the histories of many southern families were “white-washed” in the later nineteenth and twentieth centuries.

I will look for your additional articles once this one is published; I hope you will consolidate them into a single volume at some point as it would make interesting reading.  Not having read your other pieces, but reading in between the lines of this essay, I suspected the Lumbee and possible Melungeon peoples of VA/NC might figure into this.  I appreciate the details and examples you provided.

I agree that universal autosomal DNA testing would be enlightening, particularly for people born pre-1940 in terms of solving genealogical mysteries and identifications and for younger people in terms of revealing some of the genetic complexities of their own pasts as well.  Using autosomal DNA, I have been able to document shared segments present in people born in the 1960s/1970s that date from shared ancestors who lived in the middle 17th century.  I have tested a number of people born 1910-1930 and find this to be even more the case.  Within my own immediate family, we have small but identifiable elements of sub-Saharan African DNA as well as Middle Eastern (Iraq/Iran), Basque, and North African/Moroccan, and European Jewish ancestry.  It is possible to trace the widely shared sub-Saharan African segments to a “white” ancestor who, with several other mixed-race families, signed a NC petition opposing taxation for free blacks in the 1760s and I have good ideas as to the lines through which some of these other DNA elements were inherited.  I very much agree with your premise there about what DNA can teach us about ourselves, our society, and our real history. 

It would be interesting should evidence of crypto-Jewish practice turn up in New England; I agree with your softening the conjecture as you mention. 

There is clear evidence of this sort of activity in colonial Spanish settlements, as well as traditions (similar to some of the ones you mention) in families.  I teach in a predominately Hispanic region and remember being told by students twenty years ago when I started teaching here that their Garza ancestors were crypto-Jews who came to Mexico in the 1500s and had a student last year who told me of his dying grandfather’s revelation of not only Jewish ancestry but hidden Jewish practice in his family (my student subsequently “converted” to the Jewish faith although reared nominally Catholic).  Every once in a while I hear similar tales of crypto Jewish ancestry, and just recently we had a graduate student working on a project to identify families throughout the region with Jewish heritage.

Good point about the timeline for crypto-Jews.  I was thinking about more prominent families like the Devereux family, though, who, if they were of Jewish heritage, must have hidden or suppressed the fact prior to 1290.  But you are correct that 1290-1620 is a much more manageable time frame for knowledge about earlier to exist and possibly for cultural elements to persist.

Thanks for the additional context and explanation you provided on the evolution of the overall argument and the decision to focus on Madison Grant so closely in some sections.  This makes sense.

I like the changes you made.  I think they do improve the flow and address the concerns I originally had.

As I noted in the previous comments, I think this is very interesting and important research with a great deal of potential.  DNA offers tremendous potential to understand who we are as individuals and collectivities for those willing to unlock it, and I am glad you are making this effort.

A few little notes, mostly relating to adding punctuation to the recent changes:

Line 156:  Add period at end of sentence/paragraph.

Lines 236-238:  Glad you added the reference to multiple donors.

Not particularly relevant here, but it’s worth remembering that people sometimes identify the wrong ancestor.  With my own surname, there was a prominent individual “Captain X Y” who served as a longtime county justice in Virginia in the 1600s and served multiple terms in the House of Burgesses.  Numerous different families claim him (wrongly; everybody wants him) as their ancestor, with the result that he is listed as paternal ancestor in three separate and unrelated (at least within several centuries; if a connection exists, it was likely 1400s or earlier) lineage groups within the same Y-DNA Surname Study.  As it turns out, I can prove that he was the ancestor of none of them, including my own family (although several of my Y-DNA matches list him as ancestor).  I believe I have identified one male line of descent from him, but no one in that lineage has tested!  So, despite what the surname study says, there is still no bona fide DNA from “Captain X Y” to compare with these three lineage groups to determine which, if any, may be related to “Captain X Y” and his family.

Line 260:  Good addition

Line 56, Section 4.6, page 12:  It would be interesting to know if this is the line of Rev. Stephen Batchelor/Bachiler.  The Bachiler/Batchelder DNA Project does not seem to contain lineage summaries for members.  So far as I am aware, despite some pretty careful studies, Rev. Stephen’s paternity and background has never been proved prior to his matriculation at St. John’s College, Oxford, in the 1580s.  This is not particularly relevant to this study, but I have a personal interest in Bachiler.

Line 245, 252:  Don’t forget parentheses around Goss examples.

Line 414-415:  Don’t forget parentheses for Talmadge example.

I think the article is fine and ready for publication but would advise briefly adding the parentheses and the missing period before final publication.

Author Response

Dear Reviewer 1, Thanks again for your support!

I strongly suspect we are cousins from Virginia. Our autosomal ethnic content is virtually the same -- except I have traces of South Asian (likely Gypsy/Roma), as well. It was the autosomal testing that stimulated some of the research conducted over the past few years. But mostly it was finding out that my paternal surname was incorrect; the real paternal ancestor turned out to be a Jamestown colonist, so now I can join the Jamestowne Society (that is, if they are letting-in illegitimate descendants...). This happened during the Civil War, apparently -- another big source of confused paternity in the South.

    I made the edits you suggested, except for the period needed in line 156 (my computer would not let me search line numbers).

     The Atlantic Creoles term seems really useful and interesting. I will look into it;. If autosomal DNA samples could be organized into a large data set from people who can document their ancestry back to the 1500s-1700s, it would be an excellent addition to this whole line of research!